# Chronoradiobiology of Breast Cancer: The Time Is Now to Link Circadian Rhythm and Radiation Biology

**DOI:** 10.3390/ijms23031331

**Published:** 2022-01-25

**Authors:** Nicolas Nelson, Joseph Lombardo, Lauren Matlack, Alexandria Smith, Kamryn Hines, Wenyin Shi, Nicole L. Simone

**Affiliations:** Sidney Kimmel Cancer Center, Department of Radiation Oncology, Thomas Jefferson University, Philadelphia, PA 19107, USA; ngn001@students.jefferson.edu (N.N.); Joseph.Lombardo@jefferson.edu (J.L.); matlac42@rowan.edu (L.M.); as7589@pcom.edu (A.S.); Kamryn.Hines@jefferson.edu (K.H.); Wenyin.Shi@Jefferson.edu (W.S.)

**Keywords:** breast cancer, chronobiology, circadian rhythms, radiation, chronoradiotherapy

## Abstract

Circadian disruption has been linked to cancer development, progression, and radiation response. Clinical evidence to date shows that circadian genetic variation and time of treatment affect radiation response and toxicity for women with breast cancer. At the molecular level, there is interplay between circadian clock regulators such as PER1, which mediates ATM and p53-mediated cell cycle gating and apoptosis. These molecular alterations may govern aggressive cancer phenotypes, outcomes, and radiation response. Exploiting the various circadian clock mechanisms may enhance the therapeutic index of radiation by decreasing toxicity, increasing disease control, and improving outcomes. We will review the body’s natural circadian rhythms and clock gene-regulation while exploring preclinical and clinical evidence that implicates chronobiological disruptions in the etiology of breast cancer. We will discuss radiobiological principles and the circadian regulation of DNA damage responses. Lastly, we will present potential rational therapeutic approaches that target circadian pathways to improve outcomes in breast cancer. Understanding the implications of optimal timing in cancer treatment and exploring ways to entrain circadian biology with light, diet, and chronobiological agents like melatonin may provide an avenue for enhancing the therapeutic index of radiotherapy.

## 1. Introduction

Decades of research demonstrate that radiation responses vary across an organism’s circadian period. The emerging field of chronoradiobiology examines the biological relationships between the complex mechanisms of circadian regulation and cellular radiation responses with the goal of improving the therapeutic index of radiation treatments.

Understanding circadian regulation, disruptions, and downstream effects that can impact radiation therapy could lead to potential improvements for patients. Data on circadian disruption and clock gene regulation may lead to new approaches to personalize care.

In this review, we explore the fundamentals of chronobiology, focusing on the relationship to breast cancer pathogenesis, treatments, toxicity, and outcomes. The epidemiological and molecular associations of breast cancer with circadian pathways will be discussed as well as the interplay of circadian clock genes and radiation therapy. We then propose practical methods for leveraging circadian rhythms that may someday be used in radiotherapy, with potential roles for time-restricted diets and chronopharmaceuticals.

## 2. Chronobiology

Chronobiology—the study of biological rhythms and the biomolecular clockwork that drives them—has been implicated in the initiation of human disease events like sleep disorders, stroke, myocardial infarction, and more recently cancer. Harnessing the same principles of chronobiology that link it to pathogenesis provides an innovative approach for enhancing radiotherapy by altering the molecular pathways that govern biological rhythms.

### 2.1. Circadian Rhythms

Classically, chronobiology focuses on circadian rhythms, or the periodic physiological fluctuations that cycle over approximately 24 h. These oscillations are found in all domains of life, with the purpose of synchronizing an organism’s homeostasis with the rotation of the planet [1]. In fact, mammals originally evolved as nocturnal animals with adaptations that suited them to night-time activity, but the extinction of the non-avian dinosaurs opened a niche for some species to adapt to a diurnal lifestyle [2,3].

In modern primates, circadian rhythms have been shown to regulate the expression of over 80% of the transcriptome [4], and these complex systems have been explicitly linked to key physiologic processes in humans: sleep–activity cycles [5,6] neuroendocrine axes [7,8] (Figure 1), cardiovascular function [9,10], autonomic tone [11], immunity [12,13,14], neuroplasticity [15], epigenomic rhythms [1,16], microbiome activity [17,18,19], aging [15,20] and the development of different cancers [21,22,23,24,25,26,27]. Importantly, circadian regulators form a strong, bidirectional relationship with metabolism; chronobiology modulates cell metabolism and nutrient preference, and food intake feeds back into circadian rhythms through a number of modalities including nutrient availability and changes in oxidative status [1,28,29,30,31].

### 2.2. Clock Genes

These physiologic oscillations are driven by a highly conserved set of clock genes that form an autoregulatory transcription–translation feedback loop. The classical model of this core clock network is illustrated in Figure 2. For consistency, some common protein aliases will be used throughout this review, i.e., ARNTL will be referred to as “BMAL1”, and NR1D1 will be referred to as “REV-ERBα”. The heterodimer BMAL1:CLOCK binds E-box DNA response elements and enhances the transcription of clock-controlled genes, which include other clock proteins, i.e., isoforms of PER, CRY, REV-ERB, and ROR. In addition to their numerous regulatory activities in the cytosol (including DNA damage repair and cell cycle gating), PER and CRY proteins heterodimerize in the nucleus to suppress BMAL1:CLOCK activity, inhibiting their own transcription until their levels decrease as they degrade over time. This ~24-h cycle is reinforced by other clock proteins; in addition to their respective extranuclear roles, RORα increases transcription by binding the ROR response element (RRE) in the BMAL1 promoter, and REV-ERBα suppresses it [16]. Other clock-controlled gene products that serve as circadian regulators include BHLHE40 and BHLHE41, often referred to as “DEC1” and “DEC2”, respectively; in addition to other functions in the cytosol, they bind the E-box promoter and prevent their own transcription [32].

These proteins enforce their regulatory effects through a host of known mechanisms: regulating the activity of transcription factors, conditionally dimerizing with different partners, binding enzymes to modulate activity, and facilitating posttranslational modifications like phosphorylation and acetylation [16]. For simplicity, core clock proteins are often grouped into a “positive limb” that drives the clock forward (BMAL1, CLOCK, RORα) and a “negative limb” that opposes it (PER1, PER2, PER3, CRY1, CRY2, REV-ERBα, DEC1, DEC2) (Table 1).

All nucleated somatic cells exhibit self-sustaining circadian rhythms that emerge from this biomolecular clockwork [39]. Although circadian rhythms are distinct from the cell cycle, clock proteins can regulate the expression and activity of key players in cell cycle progression [55,56]. The core clock network has been shown to regulate circadian rhythms of quiescence [57], stemness, plasticity, and timed gating of cell cycle progression [57] in various cell types. Understanding the relationships between negative and positive regulators of the clock system is essential to understanding how to harness their benefit.

### 2.3. Hierarchical Organization and Zeitgebers

Circadian rhythms can only *approximate* a 24-h period. To prevent cells, tissues, and organs from desynchronizing into their own independent rhythms, the suprachiasmatic nuclei (SCN) of the hypothalamus acts as a central circadian pacemaker [58]. The SCN defines the whole-organism circadian phase—the physiologic time of day—which is communicated to the rest of the body through neuroendocrine signals [59]. Because the cells of the SCN are governed at the molecular level by clock proteins, natural circadian periods can be slightly longer or shorter than 24 h. Genetic and epigenetic variations in the core clock network result in an emergent characteristic known as an individual’s chronotype [60,61,62]. Faster clockwork means shorter endogenous periods, resulting in a chronotype that leads to a propensity for “morningness”; slower clockwork leads to a propensity for “eveningness” [58].

Uncorrected, these slight deviations would lead to free-running periods that are out of phase with the Earth’s rotation; however, the central pacemaker’s circadian rhythm can be realigned through circadian entrainment (Figure 3). To keep in line with the day–night cycle, the hypothalamus takes input from environmental time cues—referred to as zeitgebers or “time givers”—and uses this information to calibrate the suprachiasmatic nuclei [6]. This central pacemaker then sends signals throughout the body via different pathways that include the pineal gland’s production of the hormone melatonin [63]. This coordinates the body’s peripheral clocks, producing circadian oscillations in cell activity and organ function. These outputs in turn provide feedback to the central pacemaker in the form of secondary zeitgebers like food intake, exercise, and body temperature [28,64,65].

Light is the body’s primary zeitgeber. Bright light that contains blue wavelengths (e.g., daylight, standard electrical lighting) stimulates a nonvisual pathway from the retina to the central circadian pacemaker in the hypothalamus [5]. In addition to other alerting effects, this acutely suppresses the release of melatonin from the pineal gland [6,63].

Blue light suppresses melatonin in a dose-dependent fashion [66] while triggering other neurologic responses [6]. Timed exposures to polychromatic or blue-enriched light have been used both to advance and delay the circadian phase in humans [40,67]. Chronic exposure to light at night shifts the circadian phase later and has been shown to diminish the amplitude of melatonin released each night [59,68]. This forms the basis whereby rotating shift work or long-term exposure to artificial light at night leads to circadian disruption.

Artificial lights are used to treat major depressive disorder with seasonal pattern [69], but they may also be used to intentionally calibrate the central circadian pacemaker (see Figure 4) resulting in entrainment to a new, phase-shifted period (see Figure 3) [40,67]. Special lighting arrays are currently in use aboard the International Space Station for this purpose, to properly entrain the circadian rhythms of astronauts who do not experience 24-h day–night cycles [41,70]. In combination with proper scheduling, bright-blue light has been shown to be effective at helping combat jetlag and treating specific neuropsychiatric conditions on Earth [36].

Altogether, this maintains the harmony between body and ecosystem; the circadian phase of each organ system (i.e., their physiologic time) remains in sync with one another, and the whole system is aligned with the environment (i.e., the external time) as outlined in Figure 4. The concept of entrainment and its ability to alter circadian rhythms will be crucial for harnessing the benefits of chronobiology for therapeutic interventions.

### 2.4. Circadian Amplitude

Circadian rhythms depend on a robust circadian amplitude—the number of clock-driven proteins that are expressed, or the degree to which they oscillate over 24 h (see Figure 3). It follows that the activity of antitumor pathways that rely on clock proteins will vary with clock gene expression, posttranslational modification, and localization. Disruptions to the rhythm decrease the circadian amplitude [59,71]. To illustrate this at the biomolecular level: the function of the gamete-expressed protein PAS domain containing receptor 1 (PASD1) is to suppress the circadian amplitude, and it becomes oncogenic when expressed ectopically in somatic cells [72] by inhibiting apoptosis [73].

## 3. Circadian Disruptions and Breast Cancer

This section will delineate the importance of chronobiology in the disease progression of breast cancer by demonstrating epidemiological evidence and molecular associations.

### 3.1. Epidemiology

Researchers have characterized extensive relationships between circadian dysfunction and human cancer development, prognosis, and treatment [22,25]. In fact, shift work that involves circadian disruption has been recognized by the World Health Organization’s International Agency for Research on Cancer as a probable human carcinogen (Group 2A) beginning in 2007 [74]. Since then, many studies have corroborated the increased incidence of breast cancer in women exposed to light at night, an occupational hazard of shift work. Recent epidemiological studies are summarized in Table 2. The correlation is strongest with longer exposures to occasional night shifts (>20 years), shorter durations of continuous night shift, or when the history of shift work occurred in early adulthood.

The increased breast cancer incidence in night shift workers is likely multifactorial but may be related in part to melatonin suppression arising from exposure to artificial light at night [21]. Understanding the biological basis for the epidemiological observations represented in Table 2 may illuminate the clinical implications of circadian function.

### 3.2. Impact of Circadian Disruption on Health Disparities

Epidemiologic data suggest that Black and African American (AA) patients experience a greater burden of circadian disruptions due to social impacts and other external factors that alter levels of melatonin in the body. AA patients experience worse breast cancer outcomes even when treated with the same therapies, and circadian disruptions might be one factor in this disparity. In the United States, more AA workers perform rotating shiftwork than their Caucasian counterparts, and this disparity is expected to increase over time [42]. There are data that further explore the link between poor sleep quality and the development of triple-negative breast cancer in AA women [87]. In addition, AA patients may have a slower response to circadian phase shifts than Caucasian Americans, suggesting that the effects of circadian disruption might be longer-lived, allowing their risks to compound overtime [88]. Together, the higher rates of night-time shiftwork among AA patients and the resulting circadian and estrogen perturbations indicate further research to explain the nuances of their effects.

### 3.3. Melatonin and Breast Cancer

Melatonin, 5-methoxy-*N*-acetyltryptamine, is a naturally occurring hormone that is produced from tryptophan by the pineal gland. It is secreted in response to the environmental change from light to darkness. In humans, this helps synchronize organ systems in anticipation of the inactive or “rest” phase (see Figure 4). In all mammals, melatonin peaks in the evening, whereas cortisol peaks in the morning. In fact, the time at which melatonin begins to spike in the evening (dim-light melatonin onset, DLMO) is the current gold standard for measuring an individual’s circadian phase. Cortisol release as well is partly under the control of the suprachiasmatic nuclei [59].

Melatonin dysregulation is also linked to cancer development since it impacts anaerobic glycolysis, DNA repair, and angiogenesis [89]. Breast cancer models have shown that light-induced melatonin suppression leads to increased blood glucose and facilitates tumor cell proliferation; conversely, increased melatonin decreases the Warburg phenomenon and inhibits tumor growth [43]. Retrospective analyses have shown decreased night-time melatonin in women with estrogen receptor-positive (ER+) breast cancer and correlations between tumor size and peak level of night-time melatonin [90]. This is consistent with the murine xenograft breast cancer model in which transfusions of melatonin-rich blood significantly reduced tumor burden in comparison to transfusions from age-matched women whose serum melatonin levels had been suppressed by exposure to bright light [21,43].

Melatonin also decreases estrogen production, which may occur via the CRY-interacting protein TIMELESS that regulates sphingolipid metabolism-directed breast cancer cell growth [91].

### 3.4. Molecular Clock Dysfunction and Breast Cancer Risk

Different clock gene mutations and expression patterns have been implicated in the development of several cancers as well as poorer outcomes. With respect to breast cancer development, rhythmic clock gene expression is suppressed or obliterated in more aggressive cancer types, whereas a functional circadian clock is often retained in ER+, human epidermal growth factor receptor 2-negative (HER2-), low-grade breast cancers that have not yet metastasized [51]. This underscores the importance of a balanced circadian network; loss of positive-limb function reduces circadian amplitude, which can result in the loss of tumor-suppressing activity from the negative limb (Figure 5) [92].

Available meta-analyses of cancer patients’ clock gene expressions did not account for time of sample collection; nevertheless, differential patterns have been described. Low PER1 and PER2 expression is linked to breast cancer development and poorer outcomes [71]. Comparing breast cancer to adjacent tissue, PER1, PER2, PER3, and CRY2 levels are decreased; CLOCK is increased; and CRY1 downregulation was found to escalate directly with breast cancer stage [93]. Silencing of the negative-limb regulator DEC2, a purported intermediary between circadian rhythm and tumor progression, enhanced the viability, invasiveness, and colony-forming potential of breast cancer samples [32].

Differences in the function of the timeless circadian regulator protein (TIMELESS), an effector of the core clock, have been correlated specifically with ER+ and progesterone receptor-positive (PR+) breast cancers, i.e., this is another example where clock effector function has been linked to hormone-sensitive cancer development. Similarly, different levels of DEC1 and DEC2 mRNA were measured among breast cancer populations, with increased expression in PR+ cases and decreased expression in HER2+ cases [54]. The CRY2 genotype of breast cancer patients has also been correlated with ER status [37,44,94], and *PER3* loss is associated with recurrent ER+ tumors [95]. For each of these, future research might consider whether these decreases represent lower peak levels (i.e., a deficiency in their circadian maxima) or constitutively reduced baseline levels around the clock.

In terms of genetic predisposition, logistic regression analyses have linked different *CLOCK*, *CRY1*, and *PER2* genotypes to breast cancer risk [37], and several studies have implicated specific single-nucleotide polymorphisms [45]. Specific TIMELESS alleles have been correlated with hormone-sensitive breast cancers; furthermore, hypomethylation of the TIMELESS promoter is implicated in higher-stage breast cancers, and breast cancer has been shown to overexpress TIMELESS relative to normal breast tissue [37].

### 3.5. Breast Cancer Outcomes and Treatment Response

Clock gene expression patterns have been observed in breast cancers with different clinical features, although again it is unclear whether this reflects constitutive downregulation of certain circadian regulators or a deficiency of their rhythmic peaks. Broadly, higher expressions of PER1, PER2, PER3, and CRY2 were associated with longer metastasis-free survival, and distinct prognostic patterns were found to correlate with different changes in clock gene expression depending on ER, PR, and HER2 status [93,96].

These molecular changes may underlie the downstream effects of exposures that have been implicated in breast cancer progression. For example, animal models showed that light at night disrupted nocturnal melatonin signaling, which ultimately disinhibited the growth and metabolism of breast cancer cells [97]. Light-induced melatonin suppression has also been associated with the resistance of breast cancer xenografts to chemotherapy and tamoxifen [98,99].

There are also data suggesting that responses to radiation therapy are impacted by circadian factors. The heart, an important organ at risk during radiation for breast cancer, has been shown to be at higher risk for toxicity based on circadian disruption. Mice with disrupted circadian rhythms, either through environmental sleep disruption or genetic *Per* disruption, had more post-radiation cardiac dysfunction and increased fibrosis [100]. There are also clinical trials showing that the time radiation therapy is given impacts the outcomes [101,102], which will be further discussed in Section 6, chronoradiotherapy. The ability of therapeutic radiation to effectively treat cancer relies on the ability to overcome tumor cells’ DNA damage response and ability for repair. It is clear that circadian factors and clock genes regulate the cell cycle and therefore would have an impact on radiation treatment [103].

## 4. Circadian Regulation and DNA Damage Responses

### 4.1. Radiobiological Principles

When targeting solid tumors, radiation oncologists often take advantage of the “four Rs of radiation biology” including repair of DNA damage, redistribution of cells in the cell cycle, repopulation, and reoxygenation of hypoxic tumor areas, which have all been shown to be influenced by circadian regulation. Radiation treatments are often fractionated or given one treatment a day over a period of several weeks. The interval between each radiation dose gives the surviving tumor cells time to redistribute across the cell cycle so that a new portion of tumor cells will progress to G2/M, which is beneficial because radiation and the reactive oxygen species it produces are more lethal to cells in the G2/M phase [104]. Daily treatments also ensure that the solid tumor is allowed to reoxygenate with the blood vessel network created by the tumors, allowing for increased delivery of oxygen after each treatment, increasing the tumor’s sensitivity to the next dose while allowing normal cells in the vicinity to repair sublethal DNA damage and begin repopulating the surrounding tissue [46]. Understanding how biological alterations of circadian function can alter the “four Rs” will allow for the development of strategies that improve the therapeutic index of radiation treatments, i.e., maximizing its efficacy at killing malignant cells while minimizing its toxicity to normal tissue.

### 4.2. Cell Cycle Gating

Rodent models have long shown that mammals have differential radiation responses according to the circadian phase when they are irradiated. Mouse models have shown lower levels of DNA repair in skin cells in the morning, causing higher susceptibility to ultraviolet radiation. Even LD_50_, representing the lethal dose of X-ray irradiation in animal models, has been shown to oscillate as a function of time [47,105]. Similarly, individual cells are most sensitive to radiation in the G2/M phase of the cell cycle, which is subject to clock gene regulation and the circadian phase [23,57,106] (Figure 6).

Cells paused at the G2/M checkpoint by DNA damage responses (DDR) or other mechanisms are prevented from progressing into mitosis, keeping them from passing along mutations while increasing their radiosensitivity. Regulators of cell cycle progression are regulated in turn by components of the core clock network—i.e., positive- and negative-limb clock proteins, whose cellular activities oscillate over 24 h.

Depending on cell type, this leads to windows of opportunity for cell division and a bidirectional relationship between the circadian period and the cell cycle. For example, a stalled DNA replication fork can trigger a CRY- and TIMELESS-dependent pathway that prevents the cell from proceeding through G2/M, but mouse models have shown that this response depends on the circadian availability of CRY [110,111].

The respective networks that drive these manifold processes have been shown to meet at multiple regulatory nodes relevant to oncogenesis and cancer progression, e.g., the S-prolonging effect of DEC1 was observed to suppress growth in a breast carcinoma xenograft model [53]. A key component of the interplay between circadian rhythms and the cell cycle is the activity of BMAL1:CLOCK, which communicates between the clock network, c-MYC, and WEE1 [16,23,56]. As a result, chronobiology can influence the mitotic index of a particular tumor type, e.g., the radiosensitivity of human nasopharyngeal carcinoma xenografts has been shown to oscillate with the circadian period [106].

### 4.3. Double-Strand DNA Breaks

The circadian clock has been shown to gate several points in DNA damage response (DDR) pathways. For example, in response to the double-strand breaks in DNA caused by radiation, the DDR requires PER1 to bind ATM:CHEK2 in order for it to halt cell cycle progression and trigger p53-mediated apoptosis if the damage persists (see Figure 6). Ectopic PER1 expression in human cancer cell lines impairs malignant growth, and reduced levels of endogenous PER1 is found in human breast cancer.

PER2 has also been shown to operate both as a tumor suppressor as well as an important facilitator of the DDR. In murine models, PER2 was necessary for the radiation-induced upregulation of clock gene proteins that resulted in better tumor suppression and survival [112]. In human cells, both PER1 and PER2 have been shown to facilitate apoptotic pathways driven by the tumor suppression protein p53 [44,48,74].

Together, this indicates that, perhaps, radiation efficacy could be potentiated in certain cells during times with high levels of PER. These high-PER periods could either be predicted from a patient’s circadian phase or induced by manipulating zeitgebers like food intake.

Notably, double-stranded DNA breaks can only be repaired by enzymes that can perform homologous recombination, of which BRCA1 and BRCA2 are examples, which could suggest that proper PER function may be a last line of defense for cells bearing *BRCA* mutations. In cells that lack a functional BRCA, this would make PER perturbations especially hazardous for the stability of the genome. It is also possible that their respective DDR pathways are not entirely redundant; BRCA1 has been shown to interact with PER1 and PER2 in a yeast two-hybrid model [48,96,113], and specific PER mutations are purported to predict *BRCA* patients’ response to chemotherapy and survival [114].

### 4.4. Hypoxia Responses and Reoxygenation

Shifting the oxygenation profile is also key to helping with radiation sensitivity. Interestingly, the classical role of BMAL1:CLOCK is to promote the transcription of genes whose promoters contain the E-box element, like the gene for hypoxia-inducible factor 1α (HIF-1α),the subunit of HIF-1 whose expression is regulated by oxygen levels [25,48]. The relationship between HIF-1 activity and circadian regulation, however, is more nuanced than “positive-limb proteins promote, negative-limb proteins inhibit”; for example, PER2 recruits HIF-1 to its target genes [115]. Though beyond the scope of this review, circadian and HIF pathways constitute yet another bidirectional relationship, in this case serving to regulate metabolic adaptations to low oxygen levels [30].

It has been suggested that any circadian reprogramming that leads to an overexpression of HIF-1α can open a path for malignant transformation. HIF-1α is overexpressed in tumor cells, enabling several adaptations like the upregulation of angiogenic factors and glycolytic enzymes to maintain ATP production in the absence of more complex pathways like fatty acid oxidation. Shen et al. note that, because circadian dysrhythmia is known to increase the radiosensitivity of healthy tissue, the altered clock networks of tumor cells in HIF-1α–driven cancers may make them more sensitive to radiation than the healthy tissues surrounding them [25].

## 5. Metabolic Circadian Entrainment and Regulation

### 5.1. Epidemiologic Data Linking Metabolism and Circadian Dysregulation

Desynchronization of circadian rhythms have also been implicated in the etiology of metabolic diseases like obesity, type 2 diabetes, and cardiovascular disease [31,58,116,117,118]. People who are obese or diagnosed with type 2 diabetes have more circadian dysfunction, potentially leading to an increased incidence of cancer or progression of disease. Changing the time that a person eats or the composition of their diet can alter and decouple the peripheral circadian clock. It has been shown that shift work is linked to higher risk of both obesity and diabetes [119].

### 5.2. Energy Sensing

The circadian phase modulates dietary processes, from an organism’s food-seeking behavior all the way down to the nutrient preference of an individual cell. Food intake can act as a secondary zeitgeber to modulate cells’ circadian phase. Circadian rhythms evolved in order to encourage feeding and anticipate nutrient availability during the active phase [1].

For example, BMAL1 activity increases the availability of NAD^+^ which ultimately activates liver enzymes that are involved in fatty acid oxidation, increasing ATP production, and decreasing the preference for monosaccharides as a nutrient source. Mice without functional *BMAL1* genes are deficient in this pathway, and they also demonstrate impaired hunger drive which provides negative feedback against BMAL1:CLOCK activity [28,120].

One notable core clock effector is the nocturnin protein (NOCT), whose level of expression fluctuates throughout the day and communicates between the circadian phase and lipid metabolism [121]. NOCT governs nutrient use by regulating the transcripts of proteins necessary for mitochondrial function and the citric acid cycle [122], and it can modulate a cell’s NAD^+^ availability without affecting its redox status or NAD^+^:NADH ratio [123]. Finally, when the core clock network initiates CRY1 destabilization, the metabolic pathway driven by AMPK is directly affected [49,50].

These are only a few key examples that demonstrate how metabolic feedback systems are braided directly into the circadian network.

### 5.3. Dietary Modulation

Because nutrient availability is known to entrain circadian phases and modulate metabolic pathways, it stands to reason that dietary modification could be used to affect cancer outcomes. Clinical trials of dietary modulation—caloric restriction, intermittent fasting/time-restricted feeding, and carbohydrate restriction/ketogenic diet—have been of interest to oncologists, and several have resulted in promising results for different cancer populations [124,125]. Restricting food intake to designated time windows has been shown to dramatically reduce serum growth hormone, leptin, and insulin, increasing insulin sensitivity three-fold and selectively rendering tumor cells more susceptible to cytotoxic therapies. In fact, caloric restriction has been shown to enhance radiotherapy for triple-negative breast cancer [126] by increasing tumor control and decreasing metastasis [127,128]. Syngeneic animal and in vitro models have suggested that the synergetic effect happens in a phase-dependent fashion, with radiation given in the nutrient-deprived phase [126].

Supplementing radiation treatment with dietary modulation strategies like time-restricted feeding may improve tumor control by regulating circadian functions. These periods of nutrient deprivation or fasting may in fact mediate their antitumor effects by using the same machinery that allows food intake to act as a secondary zeitgeber, e.g., caloric restriction drives an oxygen-dependent cell environment via pathways that use core clock proteins and other circadian effectors. Although beyond the scope of this review, the microbiome is also an important consideration; gut microbiota have been proposed as a mediator of circadian radiosensitivity [17].

Understanding the biological effects of fasting on circadian function may allow for optimizing radiation response. Since food intake is an important secondary zeitgeber, it follows that a scheduled “zeitgeber diet” could potentiate clock entraining [28,64,121].

## 6. Leveraging Circadian Rhythms for Therapeutic Benefit

### 6.1. Chronoradiotherapy

The emerging field of chronoradiotherapy examines biological relationships between circadian regulation and cellular radiation responses in order to improve the therapeutic index of radiation. To date, there is limited preclinical and clinical data that suggest that altering circadian mechanisms could be used to improve outcomes. Preclinical evidence demonstrates that an organism’s response to radiation will vary across its circadian period, i.e., model animals demonstrate circadian radioresistance and radiosensitivity [129]. In human xenograft models, chrono-modulated radiotherapy was noted to improve tumor control, and it demonstrated a synergistic effect with other cytotoxic therapies [106].

Although there is little data on the use of primary zeitgebers to alter the radiation response, there are data demonstrating that secondary zeitgebers such as diet may alter the molecular milieu to improve radiation response. Due to the bidirectional relationships between circadian phase, metabolism, and adipocyte activity, it is worth investigating the extent that chronobiology might underlie the preliminary success of interventions like caloric restriction and time-restricted feeding for improving radiotherapy outcomes.

To date, at least seventeen clinical studies have demonstrated that the time of radiation can decrease toxicity and improve local control and overall survival [60,130,131] (studies on non-breast cancers are outlined in the Appendix A). The current data are overall limited and include varying results, but the findings may generate hypotheses for further research. Importantly, none of these studies utilized biomarkers or questionnaires to identify patients’ individual circadian phases at the time of their treatments, i.e., they used external time as a proxy for physiologic time. Table 3 outlines the two breast-cancer-specific studies published to date; neither of them, however, constitute actionable clinical recommendations at this time.

The retrospective breast cancer study suggested that patients who received doses after 15:00 h had a higher incidence of grade 2 or higher, acute skin toxicity than patients treated before 10:00 h [102]. This seems to contrast the results of the prospective trial, which found that radiation before 12:00 h increased the rate of acute breast erythema versus radiation after 12:00 h [101]. The latter study’s preference for later radiation administration is more consistent with the temporal radiotoxicity profiles of cervical, rectal, and esophageal cancer treatments (see Appendix A); however, the temporal separation between groups in the retrospective study should have been better equipped to detect the effect of time of day, assuming each patient had an ideal, eurhythmic circadian rhythm (see Figure 3).

Although there is a paucity of molecular epidemiological data to validate hypotheses, it has been noted that a structural variant of PER3 was associated with the incidence of breast cancer in young women [82] and those that have PER3 variants also have a higher burden of long-term toxicity of treatment. The time-dependence of delayed breast toxicity after radiation seems to depend on PER3 and NOCT alleles. The increased incidence of late erythema for the morning group in the study by Johnson et al. was shown to depend on patients’ genotype (*p* = 0.03), i.e., a single-nucleotide polymorphism in NOCT—a link between circadian rhythms and metabolism—and a variable-number tandem repeat in PER3 [101]. This is yet another example where the status of clock genes has been associated with treatment response, suggesting that future studies should include chronobiologic data like patient chronotypes, as they may impact the results [129].

As more data are collected, attention must be paid to what effects are being studied in each study design, appreciating that effects will likely vary from tissue to tissue. For example, hormone-sensitive cancers may respond differently to chronobiological regulation. One retrospective review found that the time-dependent improvement in response rates for palliative bone irradiation was only observed in female patients, which Chan et al. speculated may be related to differential ratios of sex hormones [27]. It is also important to note that some radiation modalities and fractionation regimens have different effects at the cellular level, and they might therefore be expected to interface with circadian clock effectors differently, e.g., the role of PER1 in double-strand DNA breaks.

### 6.2. Chronopharmaceuticals

Of the 100 top-selling drugs in the United States in 2014, 56% specifically targeted the product of a circadian gene [58]. There is growing interest in how chronobiology affects a patient’s response to medications and how medications can alter the core clock network [132,133,134], particularly in the context of breast cancer (see Table 1). Additionally, circadian rhythms have been known to impact pharmacokinetics significantly [26,114]. This is especially true for agents that act on circadian hormone receptors, e.g., glucocorticoids are more effective if given in the morning, when the body is prepared to receive signaling from the time-dependent spike in endogenous cortisol levels [3].

Extensively researched and detailed by the National Institutes of Health, melatonin remains the only hormone available over the counter as a dietary supplement. If taken orally, serum melatonin levels peak in approximately 1 h after ingestion. Taking exogenous melatonin in the morning will shift the circadian phase later, and melatonin at night shifts it earlier [135]. Complementing this regimen with timed lighting leads to a more profound phase shift: light in the morning and melatonin at night can shift the clock 1.5–2.5 h earlier per day; melatonin in the morning and light at night can shift the clock up to 2.5–3.5 h later per day. Properly timed, exogenous melatonin has been shown to decrease the latency of sleep onset and increase sleep efficiency, especially in patients with a circadian offset or a primary sleep disorder [136]. This suggests that clinical trials that use timed melatonin or prescription melatonergic drugs like tasimelteon and ramelteon for circadian entrainment may also be beneficial for patients with radiotherapy-induced fatigue [137].

There are a few human trials evaluating melatonin as an intervention to decrease the side effects of breast cancer treatment. A phase II trial found that a melatonin emulsion significantly reduced radiation dermatitis [138]. Another prospective phase II trial for women with metastatic breast cancer showed that melatonin improved both subjective and objective sleep quality [139,140]. Additionally, outcomes from various trials have shown improvements in the levels of depression and fatigue in breast cancer patients [141]. Considering the prevalence of comorbid depression among cancer patients, it is worth noting that, of the newer antidepressants researched in a Lancet meta-analysis, the melatonergic agent agomelatine ranked among the most effective and best tolerated, tantamount to fluoxetine [142].

Despite provocative preclinical data and the retrospective and prospective reviews linking circadian disruptions to breast cancer risk, there remains a sparsity of clinical data investigating melatonin as a therapeutic intervention, but research is underway. Though better known for its role as a radioprotective antioxidant, melatonin induces radiosensitization in tumor cells [143]. In vitro studies have shown that melatonin can act synergistically with tamoxifen and aromatase inhibitors [144,145]. In one clinical trial in women with metastatic hormone receptor-negative breast cancer who were no longer eligible for further chemotherapy, patients were randomized to tamoxifen alone versus tamoxifen with melatonin. Partial response rates and one-year survival were significantly higher in the melatonin adjunct group [146]. Interestingly, there has also been preclinical experimentation with novel melatonin-tamoxifen conjugate drugs [147].

Although the field is still in its infancy, combining cytotoxic therapy with pharmaceutical interventions that alter the circadian phase or the function of clock proteins has the possibility of enhancing therapeutic indices, increasing tolerability, and improving breast cancer outcomes.

## 7. Rational Circadian and Metabolic Interventions

### 7.1. Limitations of Past Research

There are important limitations to the existing body of knowledge with regards to how circadian rhythms affect radiation for breast cancer and how entrainment might be used to improve cancer outcomes. There is a paucity of research on chronobiology in clinical medicine, with less than 1% of ongoing clinical trials incorporating time-of-day considerations [52,148].

Several retrospective clinical studies have found associations between time of radiation delivery and outcomes, and few have been prospective in nature (see Table 3 and Appendix A). Although individual studies demonstrated statistical significance, study designs varied widely, and the differences in how patients were grouped presents a major obstacle to forming a consensus of strong conclusions [131]. It has been suggested that future time-of-day studies should compare groups of patients who received radiation within different narrow time windows that are separated by a few hours [149], rather than dividing groups by arbitrary cutoff times. This would ensure consistent differences in the timing of doses between patients of different groups. Furthermore, clinical radiotherapy data only exist for typical work hours, but it is possible that optimal treatment times occur overnight; bone marrow radiotoxicity is milder during the rest phase of eurhythmic mice, and circulating levels of innate and adaptive immune cells peak at different points of the night in humans [12,150].

Choosing homogeneous study populations may better resolve any time-dependent differences in the therapeutic index. In the setting of breast cancer, data support the notion that histology and breast cancer subtype may influence the constellations of clock gene changes, which would need to be studied to harness the potential of chronoradiobiology. Another limitation in the case of chronoradiotherapy is that the radiation modality being used may affect outcomes because of the core clock’s involvement in specific aspects of DNA damage repair. The considerations that could be included in future reviews and clinical studies are listed in Table 4.

Prior studies have not measured patients’ circadian phase at the time of their radiation (see Appendix A) but rather have only used external time as a proxy for internal time, which is not an accurate representation, particularly in patients with circadian dysrhythmia [151]. Studies can either evaluate circadian phase in a sleep lab or approximate it using questionnaires or clock gene expression [60,152]. Most studies have been retrospective reviews, in which these measurements are unobtainable.

Moreover, no radiation study has considered patients’ chronotypes—their “morningness” or “eveningness”. Chronotype was in fact shown to correlate with chemotherapy toxicity in women treated for breast cancer. If quantified with a questionnaire, an individual’s chronotype could also be used to calculate their expected circadian phase at a given time of day [153,154], for research purposes or for optimal treatment timing.

### 7.2. Future Directions

Optimizing chronoradiotherapy could provide innovative adjuvant treatment solutions to improve cancer outcomes for our patients. The first step toward true chronoradiotherapy in the clinic would be to have a consensus on the optimal “time of day” for treating a specific disease, e.g., early-stage triple-negative breast cancer. This “time of day” really refers to the circadian phase that would optimize the therapeutic index of radiotherapy.

The search for this optimal phase entails important caveats. The hour of maximal tumor radiosensitivity might not be the hour of maximal radioresilience for healthy tissue; if they differ, it will be crucial to determine which is more clinically relevant. Indeed, one might expect tumors to desynchronize from an individual’s circadian rhythm, considering that malignant cells often have dysfunctional clock gene expressions. A recent prostate cancer model demonstrated tumor behavior that was in phase with the circadian rhythm of host mice, but desynchronization into non-24-h rhythms has been documented in a variety of human cancers [155,156]. Future research might consider to what extent tumors resynchronize in response to host entrainment.

To ensure reproducible results of clinical studies, researchers would have to measure patients’ circadian phase at the time of their treatments, rather than relying on the time of day. We have discussed the use of dim-light melatonin onset to define the beginning/end of a person’s circadian period. The timing of the morning spike in cortisol and sleep questionnaires are other options, but none of these can measure the circadian phase at any desired instant. Other proposed alternatives include heat-based sensors to track core body temperature, heart rate variability monitors, actigraphy watches to track sleep–activity data, and other sensor-based technologies [58,151]. Peripheral clock gene expression has also been proposed, in which case samples could be collected just before radiation treatment, from blood or possibly hair [152].

However, even if researchers are able to arrive at a consensus about the ideal time window for radiotherapy for a given type of breast cancer, it is not practical to treat every patient at the same time of day; furthermore, we have seen that external time does not always line up with patients’ internal time. This fact may obscure our interpretation of the abovementioned time-of-day studies, but clinically we can use it to our advantage.

Future studies, using low-cost, low-risk strategies for entraining a person’s circadian rhythm to a desired phase, exemplified in Figure 7, and coupled with standard radiation, could improve radiation response. Bright blue light and exogenous melatonin are known to shift the circadian phase. In addition, scheduled feeding has shown even greater efficacy than melatonin supplementation in a rodent model that compared the two methods of entrainment [157,158].

One day, molecular imaging may be able to detect clock phase, and machine learning could be coupled with radiomics mapping to enhance radiotherapy dose painting [48]. These highly technical developments may arise in the future; however, given the potential benefits of chronoradiotherapy for breast cancer, there is value in working toward a technologically simpler intervention that could be safely implemented in clinical trials and eventually implemented at facilities with fewer resources.

### 7.3. Zeitgeber Diet

Once a consensus has been reached on an optimal circadian phase for treating a given disease with a particular modality, the objective would be to make a patient’s internal time align with that phase at the time of their scheduled radiation treatment (see Figure 7). To entrain circadian rhythms for optimal radiation therapy, clinicians could prescribe a set of benign interventions that work together to manipulate the circadian phase while synergizing their beneficial effects, illustrated in Figure 8.

The strategies used to entrain an optimal circadian rhythm may offer additional benefits for cancer patients. A pilot trial showed bright light therapy to reduce cancer-related fatigue and depression [141]. For patients with a non-24-h rhythm, animal models suggest that REV-ERB-targeting agents may provide a promising option for modulating period lengths while also counteracting diet-induced weight gain [160].

Melatonin reinforces bright blue light-based entrainment, but light can override the circadian effects of exogenous melatonin; in this proposed protocol, bright light should be avoided when not indicated. We have discussed the potential anticancer effects of melatonin itself, including roles in breast cancer prevention and treatment as well as amelioration of breast cancer-associated depressive and sleep symptoms [139,140]. Though not as well studied in the context of breast cancer, other melatonergic drugs exist and may have utility in this space, as well as other clock-acting agents like stenabolic (SR9009) [161].

Recently, dietary modifications like caloric restriction have been shown to improve cancer care outcomes and enhance the effect of radiation, particularly in the notoriously aggressive triple-negative breast cancer [124,125]. For the zeitgeber diet, it would be helpful to determine whether some foods are stronger zeitgebers than others.

By planning strategically timed windows of fasting, dietary restriction itself would act to reinforce the melatonergic regimen, while the circadian-entraining aspect of timed feeding could be used to potentiate the effect of timed lighting (e.g., see Appendix A). Together, we would expect these benign interventions to work in synergy, decreasing radiation toxicity while sensitizing breast tumors.

## 8. Conclusions

Understanding the interplay between chronobiology and radiobiology can lead to innovative therapies, which could be applied to improve radiation treatment response. The purpose and organization of circadian rhythms and the network of clock genes that maintain them are integral to understanding the discoveries that have already been made.

Epidemiological and biomolecular evidence has linked circadian disruptions to breast cancer, with etiologies including melatonin suppression and impaired DNA damage response systems. Learning to entrain circadian function with timed interventions like intermittent fasting can induce antitumor environments and potentiate the efficacy of radiotherapy, possibly exerting their effect through circadian effectors. Future studies should include biomarkers of circadian phase and the use of zeitgebers to reinforce circadian amplitude and ensure that each patient’s circadian rhythm is shifted to a known phase at the time of their scheduled radiation (see Figure 7).

Timed lighting, chronopharmaceutical agents, and time-restricted diets are all effective zeitgebers for shifting the circadian phase, but they have never been used in combination with the goal of promoting healthy clock gene expression and priming patients for time-dependent radiation treatment (see Figure 8). We emphasize the need for basic science research to direct future clinical studies.

Therapeutic radiation is a mainstay of breast cancer treatment, and we strongly advocate for further research that might result in the inclusion of circadian entrainment to promote robust clock function, enhance the therapeutic index of radiotherapy, reduce radiation toxicity, and improve outcomes. Circadian disruption may contribute to the pathogenesis of breast malignancies, but by harnessing targeted circadian rhythm-entraining interventions, chronoradiotherapy may contribute to the development of innovative solutions.

## Figures and Tables

**Figure 1 ijms-23-01331-f001:**
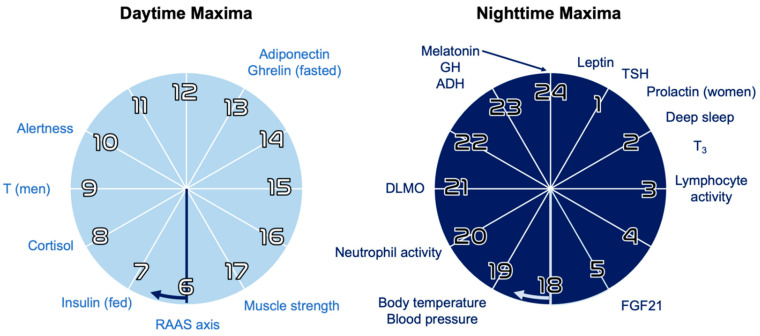
Hours of daily and nightly maxima for selected hormones and processes. These periodic oscillations are kept approximately constant via circadian rhythms of neuroendocrine signaling, which in turn is regulated by circadian clock genes [7,16]. RAAS = renin–angiotensin–aldosterone system; T = testosterone; DLMO = dim-light melatonin onset; GH = growth hormone; TSH = thyroid stimulating hormone; T_3_ = triiodothyronine; FGF = fibroblast growth factors. Adapted with permission from ref [16], copyright 2018 Springer Nature.

**Figure 2 ijms-23-01331-f002:**
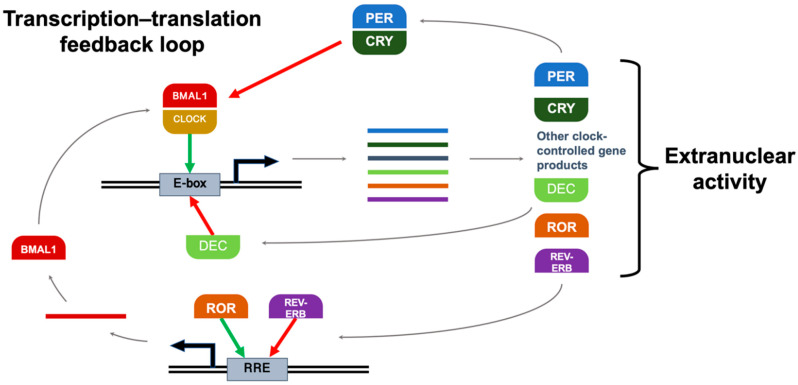
The autoregulatory feedback loop of core clock genes. BMAL1:CLOCK binds E-box and enhances clock-controlled gene transcription, including mRNA transcripts for PER, CRY, REV-ERB, and ROR. In addition to extranuclear actions, PER:CRY suppresses BMAL1:CLOCK activity, inhibiting their own transcription. This ~24-h cycle is reinforced by other clock proteins; RORα increases transcription by binding RRE in the BMAL1 promoter, and REV-ERBα suppresses it. DEC proteins bind E-box and prevent their own transcription. Green = stimulatory action; red = inhibitory action; double lines = DNA; solid lines = mRNA transcripts.

**Figure 3 ijms-23-01331-f003:**
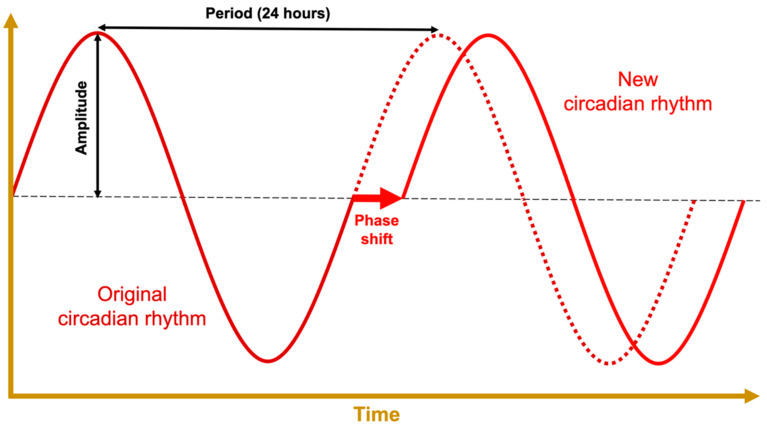
Anatomy of a prototypical waveform with relevant terminology. The vertical axis represents some circadian function, e.g., the net activity level of a clock protein. Circadian phase refers to an organism’s status along the waveform; the phase can be shifted to a new circadian rhythm through a process called circadian entrainment. Phase shifts can occur in either direction; they can be triggered by light exposure, temperature, nutrient availability, exogenous melatonin, etc.

**Figure 4 ijms-23-01331-f004:**
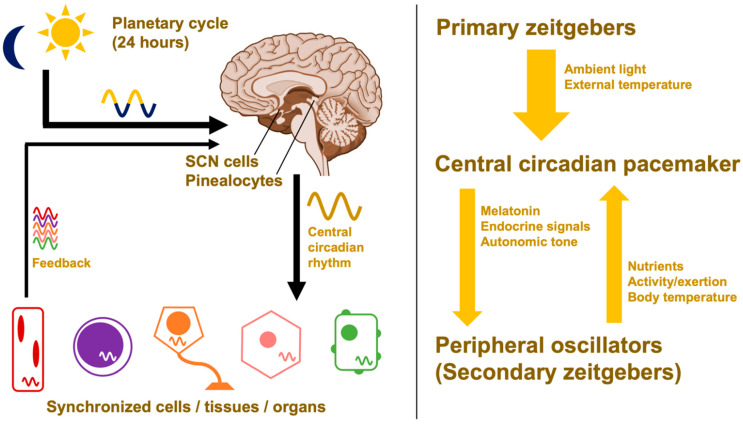
Hierarchical organization of circadian rhythms. Environmental time cues calibrate the central circadian pacemaker, which organizes regulatory structures like the hypothalamus and pineal gland, ultimately synchronizing the expression of clock genes throughout the body. At the cellular level, circadian rhythms are coordinated by the network of core clock proteins (see Figure 2). Without signals from the central pacemaker, organs and systems can uncouple into free-running rhythms. SCN = suprachiasmatic nuclei.

**Figure 5 ijms-23-01331-f005:**
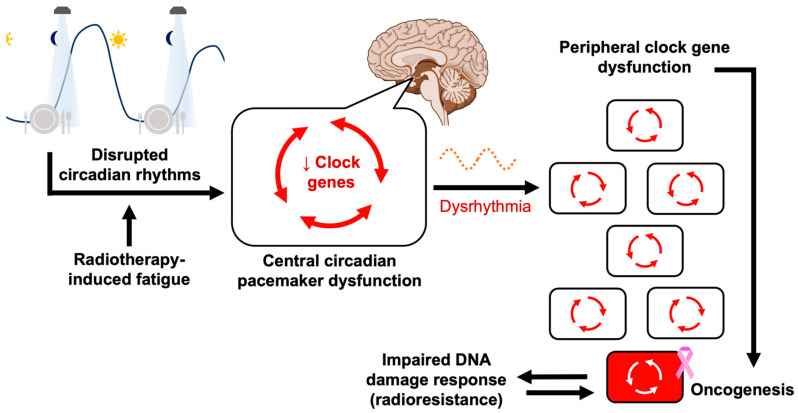
Pathway from artificial light at night to breast cancer formation. This flowchart illustrates the purported sequence of events predisposing night shift workers to breast cancer. Exposure to artificial light at night and other improperly timed cues like meals leads to circadian disruption, blunting the nightly secretion of melatonin. If the suprachiasmatic nuclei fail to integrate conflicting time signals (compromising appropriate clock gene expression), this diminishes their ability to synchronize tissues and organs, leading organ systems to develop asynchronous free-running rhythms. This inconsistent signaling can disrupt the core clock network of individual cells; clock gene dysfunction makes cells more oncogenic and tumor permissive. At the cellular level, circadian rhythms are coordinated by the network of core clock proteins (see Figure 2). See also Section 4.2 for a schematic of cell cycle gating, a key component the DNA damage response.

**Figure 6 ijms-23-01331-f006:**
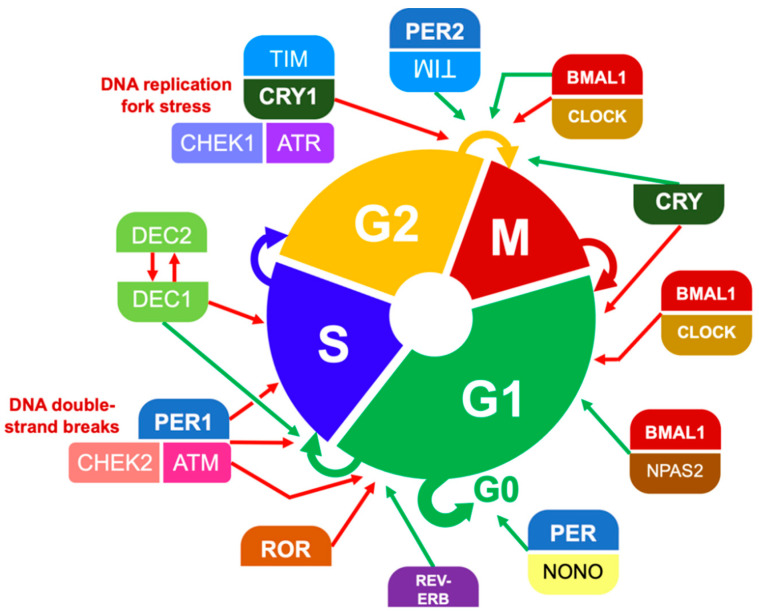
Participation of clock proteins in cell cycle gating. Simplified here are key examples of cell cycle checkpoints that depend on or are regulated by core clock proteins, whose levels and activity fluctuate over the circadian period [32,48,53,57,107,108,109,110,111]. Some cell cycle gating mechanisms are induced by DNA damage, such as CHEK2:ATM and CHEK1:ATR.

**Figure 7 ijms-23-01331-f007:**
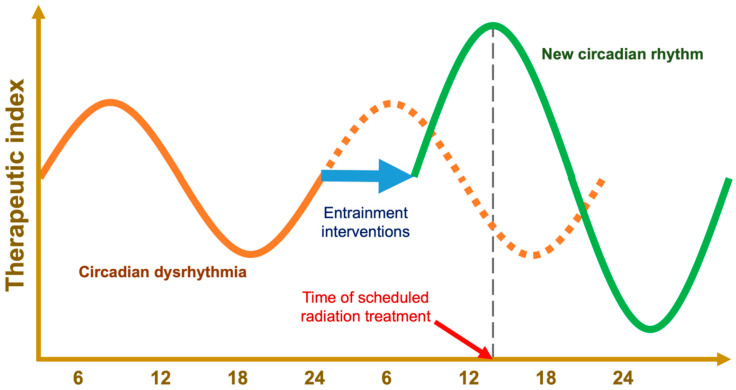
Hypothetical tracing of therapeutic index of as a function of time of day. Using zeitgebers, pathologic circadian rhythms can be entrained to an appropriate phase, a stronger amplitude, or even an altered period [159]; this provides a means for high-precision chronotherapy that does not rely on the time of day, e.g., zeitgeber-driven chronoradiotherapy.

**Figure 8 ijms-23-01331-f008:**
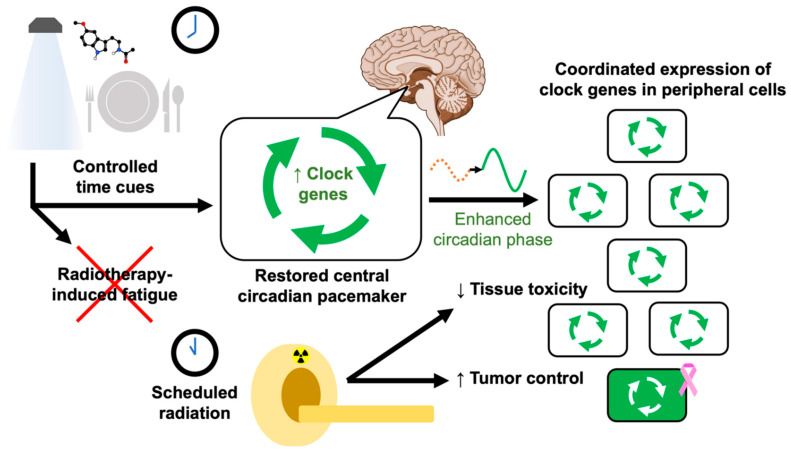
Proposed components of true chronoradiotherapy. Clinicians would start with a known optimal circadian phase for their patient’s specific pathology, i.e., the biological timepoint at which radiation will cause the most tumor damage and the least tissue toxicity. When the patient is scheduled for a radiation treatment, they are assigned a schedule of time-restricted feeding and strictly timed bright blue lighting and melatonin doses, aiming to entrain their circadian phase via properly expressed clock genes so that their optimal phase aligns with the scheduled radiation time. This would enhance the circadian amplitude of peripheral cells and synchronize them to the optimal circadian phase for the patient’s scheduled treatment time. Quality measures and research would involve confirming their circadian rhythm at each session. At the cellular level, circadian rhythms are coordinated by the network of core clock proteins (see Figure 2).

**Table 1 ijms-23-01331-t001:** Proteins of the classical core clock network. These drive circadian rhythms at the level of the cell, intersecting with multiple cancer control pathways. They are often divided into (**a**) positive and (**b**) negative limbs. For consistency, some protein aliases will be used in this review, i.e., ARNTL will be referred to as “BMAL1”, NR1D1 as “REV-ERBα”, and BHLHE40/41 as “DEC1/2”. BC = breast cancer; DDR = DNA damage response; HIF = hypoxia-inducible factor.

**a. Positive Circadian Proteins**
**Core Clock Protein**	**Most Relevant Isoforms**	**Role in Core Clock Network**	**Possible Roles in Breast Cancer**
Aryl hydrocarbon receptor nuclear translocator likeAlias: Brain and muscle ARNT-like	ARNTL (BMAL1) 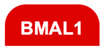	BMAL1:CLOCK binds E-box to promote transcription of clock-controlled genes 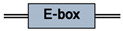	Maintains circadian amplitudeEnables hypoxia responseRegulates fatty acid oxidationDecreases with senescence[24,29,33,34]
Clock circadian regulatorAlias: Circadian locomotor output cycles kaput (mouse) homolog	CLOCK 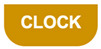	BMAL1:CLOCK binds E-box to promote transcription of clock-controlled genes 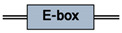	Associated with BC risk/incidenceRegulates circadian acetylationProtooncogenic c-MYC regulationSuppressor WEE1 regulationG2/M transition gating[3,27,35,36,37]
RAR related orphan receptorAlias: Retinoic acid receptor-related orphan receptor	RORA (RORα) 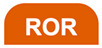	RORα binds RRE to promote transcription of BMAL1 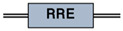	Antitumor activityAnti-inflammatory activityG1 completion gating[38,39]
**b. Negative Circadian Proteins**
**Core Clock Protein**	**Most Relevant Isoforms**	**Role in Core Clock Network**	**Possible Roles in Breast Cancer**
Period circadian regulator	PER1PER2PER3 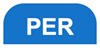	PER:CRY heterodimers suppress BMAL1:CLOCK-mediated transcription 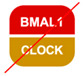	Decreased expression in BCPredict response/outcomes in BCInhibits cancer cell growthInhibits tumorigenesisRequired for ATM:CHEK2 DDRRequired for radiation-induced clock gene upregulation, DDRAdipocyte differentiationG0 cell cycle exit[3,27,39,40,41,42,43,44,45,46,47,48,49,50]
Cryptochrome circadian regulator	CRY1CRY2 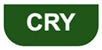	PER:CRY heterodimers suppress BMAL1:CLOCK-mediated transcription 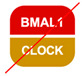	Decreases with ↑BC stageInhibits reactive tumor formationp53 Signaling, G2/M gatingER status of BCShift work, BC incidenceReduces MYC tumor burdenCounteracts hypoxia responseRequired for ATR:CHEK1 replication fork stress response[3,29,36,37,38,39,51]
Nuclear receptor subfamily 1 group D member 1Alias: REV-ERBα	NR1D1 (REV-ERBα) 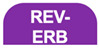	REV-ERBα binds RRE to suppress transcription of BMAL1 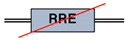	Selective lethality to BC cellsCancer cell lethalityGlucose consumptionG1/S transition enabling[3,38,39,52]
Basic helix-loop-helix family member e41/42Aliases: Differentially expressed in chondrocytes (DEC), SHARP, STRA13	BHLHE40 (DEC1)BHLHE41 (DEC2) 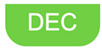	DEC is transcribed from E-box promoter, then binds E-box to prevent binding of BMAL1:CLOCK 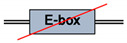	BC tumor suppression, outcomes, receptor statusDelayed S phase in BC cellsEpithelial-to-mesenchymal transition[32,53,54]

**Table 2 ijms-23-01331-t002:** Epidemiological evidence linking circadian disruption and breast cancer. Exposure to light at night, either (**a**) without or (**b**) with consideration of genotype.

**a. Human Breast Cancer Association Studies**			
**Study**	**Type**	**Exposure**	**Conclusion**
Circadian disrupting exposures and breast cancer risk: a meta-analysis [75]	Meta-analysis	Shift work, short sleep duration, employment as flight attendant	Circadian disruption is associated with an increased breast cancer risk in women. (RR = 1.14; 95% CI 1.08–1.21).
Rotating Night-Shift Work and the Risk of Breast Cancer in the Nurses’ Health Studies [76]	Two prospective cohort studies (NHS I and II)	Night shift work	Long term night shift work had a higher risk of breast cancer. Pronounced with shift work during young adulthood. (HR = 2.15, 95% CI: 1.23, 3.73
Night-shift work and breast and prostate cancer risk: updating the evidence from epidemiological studies. Night-shift work and breast and prostate cancer risk: updating the evidence from epidemiological studies [77]	Meta-analysis	Night shift work	Risk is inconclusive and more studies are required
Evaluating the Association between Artificial Light-at-Night Exposure and Breast and Prostate Cancer Risk in Spain (MCC-Spain Study) [78]	Population based multi case-control study	Artificial light at night	Prostate and breast cancer were associated with high estimated exposure to outdoor light at night
Night Shift Work and Risk of Breast Cancer in Women [79]	Case-control	Night shift work	Positively associated night shift work with breast cancer. (OR = 8.58; 95% CI: 2.19–33.8)
Outdoor light at night at residences and breast cancer risk in Canada [80]	Population based case-control study	Outdoor light at night	Outdoor light at night has a small effect or no effect on breast cancer risk
NTP Cancer Hazard Assessment Report on Night Shift Work and Light at Night [81]	Systematic review	Night shift work, light at night	Likely causal relationship of persistent night shift work, particularly in young adults, to developing breast cancer
**b. Studies Including Genetic Variants**			
**Study**	**Type**	**Exposure**	**Outcome**
Period3 structural variation: a circadian biomarker associated with breast cancer in young women [82]	Case-control study	PER3 variation	Increased risk of breast cancer in premenopausal women
Circadian genes and breast cancer susceptibility in rotating shift workers [83]	Prospective cohort	Shift work	Common variation in circadian genes play at most a small role in breast cancer risk among women of European ancestry. Neuronal PAS domain protein 2 (NPAS2) was strongly associated with breast cancer risk (*p*-value = 0.0005)
Breast cancer risk, night work, and circadian clock gene polymorphisms [84]	Population-based case-control study in France	Night shift work	Circadian clock gene variants modulate breast cancer risk. SNPs in RORA (rs1482057 and rs12914272) and in CLOCK were associated with breast cancer risk.
Circadian gene variants and breast cancer [85]	Epidemiological studies cited	Light at night	Circadian gene variants are significantly associated with breast cancer risk. BMAL1, BMAL2, CLOCK, NPAS2, CRY1, CRY2, PER1, PER3 and TIMELESS.
*BRCA1* and *BRCA2* Gene Expression: Diurnal Variability and Influence of Shift Work [86]	Cohort study	Night shift work	Lower *BRCA1* and *BRCA2* expression were found in a group of shift workers. It may be one of the potential factors related to the higher risk of breast cancer.

**Table 3 ijms-23-01331-t003:** Effect of time of day on radiation treatments for breast cancer. Overall conclusions about a preferred radiation time are indicated in bold in the findings.

Time of Radiotherapy for Breast Cancer				
Cancer Cite and Study	Type	Timing	Radiation	Endpoints and Findings
Breast Genetic Variants Predict Optimal Timing of Radiotherapy to Reduce Side-effects in Breast Cancer Patients	Prospective cohort (*n* = 343)	Before vs. after 12:00 pm (≥66% of total dose)	50 Gy in 25 fractions, 40 Gy in 15 fractions	Acute skin toxicity, late skin toxicity, clock gene alleles: Morning radiation increased acute and late breast erythema Effect of radiation time on late toxicity depended on PER3, NOCT alleles (*p* = 0.03) (Late preference)
Breast Comparison of acute skin reaction following morning versus late afternoon radiotherapy in patients with breast cancer who have undergone curative surgical resection	Retrospective (*n* = 395)	Before 10:00 am vs. after 15:00	50.4 Gy in 28 fractions before 2003, 50 Gy in 25 fractions	Acute skin reaction, survival, treatment failure: Afternoon radiation increased grade 2+ acute skin toxicity (*p* = 0.0088) No difference in treatment failure or survival outcomes (Early preference)

**Table 4 ijms-23-01331-t004:** Considerations for chronoradiotherapy. Listed are some of the factors that should be considered when grouping patients and comparing results in future chronoradiotherapy studies and clinical trials. DLMO = dim-light melatonin onset.

Patient Factors	Treatment Factors
Circadian phase, biomarkers *	Time of day, narrow ranges with significant gaps
Chronotype	Primary cancer site
Clock gene analysis	Disease stage, grade, mutations
Demographics, exposure history	Radiation dose, modality, fractionation
Radiation toxicity, tumor control	Consistency of radiation times, carryover effects
Microbiome analysis	Adjuvant/definitive therapies, immune suppression

* E.g., clock gene expression, DLMO, cortisol spike, core body temperature, heart rate variability, actigraphy, sleep questionnaires.

## Data Availability

The data that support the findings of this study are available from the corresponding author upon reasonable request.

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
