# Peer review of "Chronoradiobiology of Breast Cancer: The Time Is Now to Link Circadian Rhythm and Radiation Biology"

_ijms, 2022, doi:10.3390/ijms23031331_

Round 1

Reviewer 1 Report

Nelson, Simone, and colleagues provide a thoughtful and integrative review detailing the contributions that the circadian system (and the molecular clock mechanism) might make to therapeutic indices associated with cancer radiation treatment. They outline several convincing arguments that: 1. Circadian genes/proteins that interact to produce near 24-hour signaling patterns in cells also happen to be key regulators of cell division, apoptosis, and DNA damage responses (eg, repairs after double-strand breaks), and 2. The progression and treatment outcomes observed in various cancer types can be variably steered by circadian gene variants. While I’m highly enthusiastic about the review, I have four suggestions that I hope the authors will consider.

  1. The authors emphasize that levels of Per1, Per2, Per3, and Cry2 are decreased during the progression of breast cancer and associate with worse outcomes. This makes sense, considering the proteins’ roles in DNA repair and tumor suppression. However, can the authors contextualize the research that established these findings a bit better? All these proteins exhibit robust oscillations throughout the 24-hour day. When the authors say “decrease” do they mean that: (1) The amplitude of the oscillation is decreased (peak levels are off) or (2) Do they mean that the entire rhythm is reduced (ie, all readings done across 24 hours are lower than what control values would be)? In so many words, are levels reduced at only certain times of day or are they constitutively reduced?
  2. The authors assume that the circadian phase that will minimize radiation injury responses in normal surrounding tissue will be the same that incurs maximal tumor damage (same timepoint = best normal tissue response = worst ability of the tumor to recover). How reasonable of an assumption is this? Do we know what the phase synchrony is between the host suprachiasmatic nucleus relative to the tumor? Between the phase synchrony of certain processes in the immune system or peripheral organs (or the gut microbiota) relative to the tumor? Which processes are perfectly phase-aligned (eg, energy consumption in normal tissue is maximized/demanded at times when energy consumption is similarly demanded by tumor cells)? Which are offset in antiphase (180 degrees)? Recent work indicates that parasites can stay rhythmic in arrhythmic clock mutant hosts. Do tumors entrain to normal host tissue? What would the treatment implications be if they do or don’t?
  3. The range of the circadian cycle that the authors look to distinguish a radiation treatment difference is restricted (eg, differentiating responses across a typical 8-hour work/social day between 9am and 5pm; is morning or afternoon treatment best)? This is somewhat contrived. Are there any empirical data suggesting that treatment responses might not be better overnight, say at 2am, if the cancer patient were kept overnight in the hospital and awoken? Several aspects of the immune system are naturally elevated overnight during the habitual sleep period, including the number of circulating white blood cells. Is there any reason to not consider the possibility that tumor cells might be best targeted in the middle of the night?
  4. There are factual errors in the circadian background section. The text around line 140 implies that RHT activation reduces melatonin thereby phase-shifting endogenous rhythms. The authors make a similar comment about blue light impacting circadian phase by way of melatonin. This is not accurate. Light activation of the SCN produces three independent effects: (1) it shifts/resets the SCN’s phase (thereby shifting the pineal gland’s melatonin rhythm over the next several 24-h cycles), (2) it acutely suppresses melatonin secretion—within 5 minutes of light contact--before melatonin recovers to the level it would have been in the 2-3 hour block after the light stimulation has ended, and (3) it creates acute arousal responses that have nothing to do with melatonin. Note that there are times of night that light exposure will suppress melatonin secretion but have little impact on the phase of the melatonin rhythm. I suggest that the authors speak with a local circadian expert to unpack this text a bit better.

Author Response

The authors greatly appreciate both the enthusiasm as well as the thoughtful recommendations. We have reviewed the suggestions and observations and revised our manuscript accordingly as follows:

  1. The authors emphasize that levels of Per1, Per2, Per3, and Cry2 are decreased during the progression of breast cancer and associate with worse outcomes. This makes sense, considering the proteins’ roles in DNA repair and tumor suppression. However, can the authors contextualize the research that established these findings a bit better? All these proteins exhibit robust oscillations throughout the 24-hour day. When the authors say “decrease” do they mean that: (1) The amplitude of the oscillation is decreased (peak levels are off) or (2) Do they mean that the entire rhythm is reduced (ie, all readings done across 24 hours are lower than what control values would be)? In so many words, are levels reduced at only certain times of day or are they constitutively reduced?

We agree with the very keen point that, regarding studies of relative levels of expression of different circadian regulators, there is a key distinction between a dysfunctional circadian peak vs. a constitutively reduction. Unfortunately, to the best of the authors’ knowledge, the human meta-analyses were not able to indicate sample collection time (i.e. circadian phase). Therefore, we have attempted to clarify throughout the manuscript where measurements were made around the clock (in smaller studies and animal models). We have included the above caveats in Section 3.4:

Available meta-analyses of cancer patients’ clock gene expressions did not account for time of sample collection; nevertheless, differential patterns have been described. … For each of these, future research might consider whether these decreases represent lower peak levels (i.e., a deficiency in their circadian maxima) or constitutively reduced levels around the clock.

And in Section 3.5:

Clock gene expression patterns have been observed in breast cancers with different clinical features, although again it is unclear whether this reflects constitutive downregulation of certain circadian regulators or a deficiency of their rhythmic peaks.

2. The authors assume that the circadian phase that will minimize radiation injury responses in normal surrounding tissue will be the same that incurs maximal tumor damage (same timepoint = best normal tissue response = worst ability of the tumor to recover). How reasonable of an assumption is this? Do we know what the phase synchrony is between the host suprachiasmatic nucleus relative to the tumor? Between the phase synchrony of certain processes in the immune system or peripheral organs (or the gut microbiota) relative to the tumor? Which processes are perfectly phase-aligned (eg, energy consumption in normal tissue is maximized/demanded at times when energy consumption is similarly demanded by tumor cells)? Which are offset in antiphase (180 degrees)? Recent work indicates that parasites can stay rhythmic in arrhythmic clock mutant hosts. Do tumors entrain to normal host tissue? What would the treatment implications be if they do or don’t?

The authors agree that the original manuscript implied that maximal tumor radiosensitivity would coincide with maximal healthy tissue radioresilience. We have resolved this discrepancy with commentary and brief preliminary evidence for and against the synchronicity of human tumors in Section 7.2:

The search for this optimal phase entails important caveats. The hour of maximal tumor radiosensitivity might not be the hour of maximal radioresilience for healthy tissue; if they differ, it will be crucial to determine which is more clinically relevant. Indeed, one might expect tumors to desynchronize from an individual’s circadian rhythm, considering that malignant cells often have dysfunctional clock gene expression. A recent prostate cancer model demonstrated tumor behavior that was in phase with the circadian rhythm of host mice [Zhu 2021], but desynchronization into non-24-hour rhythms has been measured in a variety of human cancers [Klevecz 1991]. Future research might consider to what extent tumors resynchronize in response to host entrainment (see Figure 8).

A more detailed discussion of phase alignment among major biological processes would be fascinating and eventually clinically useful; however, a proper discussion of these and the relationships between circadian rhythms, immune function, and microbiota would entail separate reviews. We have made revisions to acknowledge these limitations in scope while underscoring the importance of these topics, including in Section 5.3:

Although beyond the scope of this review, the microbiome is also an important consideration; gut microbiota have been proposed as a mediator of circadian radiosensitivity [Cui 2016].

3. The range of the circadian cycle that the authors look to distinguish a radiation treatment difference is restricted (eg, differentiating responses across a typical 8-hour work/social day between 9am and 5pm; is morning or afternoon treatment best)? This is somewhat contrived. Are there any empirical data suggesting that treatment responses might not be better overnight, say at 2am, if the cancer patient were kept overnight in the hospital and awoken? Several aspects of the immune system are naturally elevated overnight during the habitual sleep period, including the number of circulating white blood cells. Is there any reason to not consider the possibility that tumor cells might be best targeted in the middle of the night?

The authors agree with the contrived nature of the existing human data to date due to the logistics of standard care. We have commented on this and the possibility that a purported optimal treatment phase might occur overnight, with one preclinical example of decreased toxicity in the rest phase. It would be reasonable for the circadian pattern of immune cell activity/localization to affect radiation responses in a clinically meaningful way, but due to the scope of the manuscript, we have limited discussion to earlier allusions throughout and the addition to Section 7.1:

… rather than dividing groups by arbitrary cutoff times. This would ensure consistent differences in the timing of doses between patients of different groups. Furthermore, clinical radiotherapy data only exists for typical work hours, but it is possible that optimal treatment times occur overnight; bone marrow radiotoxicity is milder during the rest phase of eurhythmic mice [Haus 1974], and circulating levels of innate and adaptive immune cells peak at different points of the night in humans [12].

4. There are factual errors in the circadian background section. The text around line 140 implies that RHT activation reduces melatonin thereby phase-shifting endogenous rhythms. The authors make a similar comment about blue light impacting circadian phase by way of melatonin. This is not accurate. Light activation of the SCN produces three independent effects: (1) it shifts/resets the SCN’s phase (thereby shifting the pineal gland’s melatonin rhythm over the next several 24-h cycles), (2) it acutely suppresses melatonin secretion—within 5 minutes of light contact--before melatonin recovers to the level it would have been in the 2-3 hour block after the light stimulation has ended, and (3) it creates acute arousal responses that have nothing to do with melatonin. Note that there are times of night that light exposure will suppress melatonin secretion but have little impact on the phase of the melatonin rhythm. I suggest that the authors speak with a local circadian expert to unpack this text a bit better.

In the original manuscript, the authors did imply in error that melatonin suppression is the mechanism by which light exposure contributes to circadian phase shifts. We have revised Section 2.3 in order to remove this implied causality throughout, also noting the acuity of melatonin suppression and alluding to the significance of time of light exposure, which is revisited in Section 7.2.

“To keep in line with the day–night cycle, the hypothalamus takes input from environmental time cues—referred to as zeitgebers or “time givers”—and uses this information to calibrate the suprachiasmatic nuclei [6]. This central pacemaker then sends signals throughout the body via different pathways that include the pineal gland’s production of the hormone melatonin [48]. This coordinates the body’s peripheral clocks, producing circadian oscillations in cell activity and organ function. These outputs in turn provide feedback to the central pacemaker in the form of secondary zeitgebers like food intake, exercise, and body temperature”

“Light is the body’s primary zeitgeber. Bright light that contains blue wavelengths (e.g., daylight, standard electrical lighting) stimulates a nonvisual pathway from the retina to the central circadian pacemaker in the hypothalamus [5]. In addition to other alerting effects, this acutely suppresses the release of melatonin from the pineal gland [6, 48].

Blue light suppresses melatonin in a dose-dependent fashion [51] while triggering other neurologic responses [6]. Timed exposures to polychromatic or blue-enriched light have been used both to advance and delay the circadian phase in humans [52, 53]. Chronic exposure to light at night shifts the circadian phase later and has been shown to diminish the amplitude of melatonin released each night [44, 54]. This forms the basis whereby rotating shift work or long-term exposure to artificial light at night leads to circadian disruption.”

Reviewer 2 Report

I have reviewed the manuscript, “Chronoradiobiology: The time is now to link circadian rhythm 2 and radiation biology” under consideration of publication in the IJMS journal and highly recommend it for publication. The manuscript is very well written and clearly explains the current understanding of circadian biology and its application for chronoradiotherapy. The figures are very well done and help to explain the subject. I have found an important reference regarding Cry1’s role in the regulation of ATR pathway that is highly relevant to this work and recommend it to cite; “Modulation of ATR-mediated DNA damage checkpoint response by cryptochrome 1 (2014), Nucleic Acids Research. Overall, I think this review will be of great use to the field and should be published without hesitation.

Author Response

The authors thank the reviewer’s close reading and enthusiasm, and we appreciate the recommendation; we have read the reference and incorporated the findings in Section 4.2 as an important example of the circadian dependence of some cell cycle gating mechanisms—one that in this case may pause the cell in a more radiosensitive state:

For example, a stalled DNA replication fork can trigger a CRY- and TIMELESS-dependent pathway that prevents the cell from proceeding through G2/M [Yang 2010], but mouse models have shown that this response depends on the circadian availability of Cry [Kang 2014].

Reviewer 3 Report

This is an outstanding review addressing the advantages of considering the timing for cancer radiation therapeutics, particularly in breast cancer.

The authors give an overview of circadian rhythms and their regulation by the clock gene and protein machinery and then describe the relationship between chronodisruption and the epidemiology of breast cancer. An extensive review of the literature on circadian regulation of several aspects of physiopathology is also presented. Possible interventions with diet and hormones such as melatonin, as well as time-related radiation, are suggested to improve breast cancer outcome.  

Please use consistent human gene and protein nomenclature according to genenames.org

Author Response

The authors thank the reviewer’s close reading and appreciate the correction; we have adopted the correct human and mouse notations where appropriate throughout, and we converted to HGNC nomenclature; we have indicated in Section 2.2 and in Table 1 the two main exceptions where ARNTL is more commonly referred to in circadian rhythm literature by the alias “BMAL1” and NR1D1 as “REV-ERBα.”

Reviewer 4 Report

Authors described chronoradiobiology. I have only one major comment.

They need to add other major clock genes DEC1/bhlhe40 and DEC2/bhlhe41 and discuss well. These molecules have circadian in tumor cells and regulates cell cycles.

Author Response

The authors appreciate the reviewer’s suggestion to include discussion of BHLHE40 and BHLHE41, as they are key players in the circadian feedback loop and have been studied in the context of tumor suppression and breast cancer. We have included detailed descriptions in Section 2.2 and Table 1, their key roles in the schematics of Figures 2 (circadian feedback loop) and 6 (cell cycle), and instances where they have been implicated in tumor phenotype and progression in Section 4.2:

The respective networks that drive these manifold processes have been shown to meet at multiple regulatory nodes relevant to oncogenesis and cancer progression, e.g., the S-prolonging effect of DEC1 was observed to suppress growth in a breast carcinoma xenograft model [Bi 2015]

As well as Section 3.4:

Differential mRNA levels of the circadian E-box inhibitors DEC1 and DEC2 have been found among breast cancer patients, with increased expression in PR+ cases and decreased expression in human epidermal growth factor receptor 2-positive (HER2+) cases [Fang 2020]

“Silencing of the negative-limb regulator DEC2, a purported intermediary between circadian rhythm and tumor progression [Sato 2016], enhanced the viability, invasiveness, and colony-forming potential of breast cancer samples [Fang 2020].”

Round 2

Reviewer 4 Report

Authors well improved and Dec involvement is fine. No more claim.